# 2-Deoxy-D-glucose couples mitochondrial DNA replication with mitochondrial fitness and promotes the selection of wild-type over mutant mitochondrial DNA

Boris Pantic[1,11], Daniel Ives[1,11], Mara Mennuni [1,11], Diego Perez-Rodriguez[1,11], Uxoa Fernandez-Pelayo[2], Amaia Lopez de Arbina[2], Mikel Muñoz-Oreja[2,3], Marina Villar-Fernandez[2], Thanh-mai Julie Dang[1], Lodovica Vergani[4], Iain G. Johnston[5], Robert D. S. Pitceathly [6], Robert McFarland[7], Michael G. Hanna[6], Robert W. Taylor [7], Ian J. Holt [1,2,8,9,10✉] & Antonella Spinazzola [1✉]

Pathological variants of human mitochondrial DNA (mtDNA) typically co-exist with wild-type molecules, but the factors driving the selection of each are not understood. Because mitochondrial fitness does not favour the propagation of functional mtDNAs in disease states, we sought to create conditions where it would be advantageous. Glucose and glutamine consumption are increased in mtDNA dysfunction, and so we targeted the use of both in cells carrying the pathogenic m.3243A>G variant with 2-Deoxy-D-glucose (2DG), or the related 5-thioglucose. Here, we show that both compounds selected wild-type over mutant mtDNA, restoring mtDNA expression and respiration. Mechanistically, 2DG selectively inhibits the replication of mutant mtDNA; and glutamine is the key target metabolite, as its withdrawal, too, suppresses mtDNA synthesis in mutant cells. Additionally, by restricting glucose utilization, 2DG supports functional mtDNAs, as glucose-fuelled respiration is critical for mtDNA replication in control cells, when glucose and glutamine are scarce. Hence, we demonstrate that mitochondrial fitness dictates metabolite preference for mtDNA replication; consequently, interventions that restrict metabolite availability can suppress pathological mtDNAs, by coupling mitochondrial fitness and replication.

[1] Department of Clinical and Movement Neurosciences, UCL Queen Square Institute of Neurology, Royal Free Campus, London NW3 2PF, UK. [2] Biodonostia Health Research Institute, 20014 San Sebastián, Spain. [3] Department of Pediatrics, Medicine and Nursing Faculty, Universidad de País Vasco, Bilbao, Spain. [4] Department of Neurosciences, University of Padova, 35128 Padova, Italy. [5] Faculty of Mathematics and Natural Sciences, University of Bergen, 5007 Bergen, Norway. [6] Department of Neuromuscular Diseases, UCL Queen Square Institute of Neurology and The National Hospital for Neurology and Neurosurgery, London WC1N 3BG, UK. [7] Wellcome Centre for Mitochondrial Research, Translational and Clinical Research Institute, Faculty of Medical Sciences Newcastle University, Newcastle upon Tyne NE2 4HH, UK. [8] IKERBASQUE, Basque Foundation for Science, 48013 Bilbao, Spain. [9] CIBERNED (Center for Networked Biomedical Research on Neurodegenerative Diseases, Ministry of Economy and Competitiveness, Institute Carlos III), 28031 Madrid, Spain. [10] Universidad de País Vasco, Barrio Sarriena s/n, 48940 Leioa, Bilbao, Spain. [11] These authors contributed equally: Boris Pantic, Daniel Ives, Mara Mennuni, Diego Perez-Rodriguez. ✉email: ian.holt@biodonostia.org; a.spinazzola@ucl.ac.uk

Mitochondrial DNA (mtDNA) is essential for energy production and mutations in this molecule cause an energy crisis, with consequent disease. Crucially, most heteroplasmic pathological mtDNA variants produce biochemical and clinical phenotypes only at very high levels[1–3]. Hence, it is not necessary to eradicate all the mutant mtDNA to restore mitochondrial function; instead, a modest decrease in mutant load should be sufficient to transition from a disease to a healthy state. Thus, a longstanding goal in the field has been to identify a means of increasing the proportion of wild-type molecules, as this could be an effective therapy.

Several lines of evidence indicate that active selection can determine heteroplasmy levels: rapid segregation of deleterious mtDNA variants occurs in human cultured cells[4–7], segregation bias is a feature of some mouse tissues[8,9], and selection against nonsynonymous mutants occurs in the mouse germline;[10] yet, the underlying mechanisms remain unclear. In the case of the most common pathological variant, m.3243A>G, which produces dysfunctional mt-tRNA$^{Leu(UUR)}$, the mutant molecules are selected in most cell lines[4,11,12], whereas A549 adenocarcinoma can occasionally select the wild-type mtDNAs[5,6]. We have previously suggested that differences in bioenergetics might account for the selection bias[11]. Others have shown that restrictive temperature conditions, by de-energizing mitochondria with mutant mtDNA, prevent the propagation of dysfunctional molecules in the fly germline[13]. Although the study highlighted that mitochondrial fitness can be harnessed to drive the selection of wild-type mtDNA, it did not explain how this might be achieved in the case of human pathological mutants.

A switch from glycolytic to oxidative metabolism has been proposed to be important for the selection of nonsynonymous variants in early germ-cell development in mammals[14]. Moreover, it is well known that ATP insufficiency due to mitochondrial dysfunction can be compensated by glycolytic metabolism (the conversion of glucose to lactate), even to the extent of meeting all the energy requirements of cells lacking a respiratory chain[11,12,15]. Hence, we inferred that restricting glycolysis should favour mitochondrial fitness, and thus promote the selection of wild-type mtDNA. On the other hand, it has been shown that cells with defective oxidative phosphorylation (OXPHOS) increase their reliance on glutamine metabolism to support energy production and cell proliferation[16]. Therefore, it might also be necessary to limit glutamine metabolism to inhibit the propagation of dysfunctional mtDNA. The glucose analogue 2-Deoxy-D-glucose (2DG) inhibits glycolysis[17] and restricts glutamine utilization[18] making it a strong candidate to select wild-type over deleterious mtDNA variants.

Here, we show that 2DG and another glucose analogue, 5-thioglucose (5TG), select wild-type mtDNA molecules in multiple cell types and restore the mitochondrial respiratory capacity. 2DG selectively inhibits the replication of the mutant mtDNA, as does glutamine restriction, but not glucose deprivation or inhibition of glycolysis. However, glucose-fuelled respiration is critical for mtDNA replication when glutamine and glucose availability is restricted in control cells. Adjusting the level of both these metabolites recapitulates the effects of 2DG on mtDNA replication and segregation. Thus, metabolite usage for mtDNA replication depends on mitochondrial fitness and it can be leveraged to favour the propagation of functional mtDNAs.

## Results

### Glucose analogues favour wild-type mtDNA molecules in multiple cell types and restore mitochondrial respiratory function.
Since increased glucose and glutamine utilization support cells with mutant mtDNA[11,12,15,16], we sought to promote

active selection of wild-type mtDNA by restricting their usage with 2DG. In stable, heteroplasmic A549 m.3243A>G cells, 2DG treatments led to a modest, but significant decrease in the proportion of mutated mtDNA (Fig. 1a), whereas glucose restriction did not alter the level of mutant molecules (Supplementary Fig. 1a). Moreover, when 2DG was withdrawn from the growth medium, m.3243A>G returned to its original level (Fig. 1b). These data suggest that the changes in heteroplasmy level are a consequence of 2DG, rather than the result of random drift to wild-type mtDNA, or selective death of cells with high mutant loads. 5TG, which represses the glycolytic flux to a similar extent to 2DG (Supplementary Fig. 1b), also decreased the mutant load in A549 cells (Fig. 1c).

Next, we tested the effect of the compounds on the same mutant mtDNA in another nuclear background (rhabdomyosarcoma, Myo.RD), which has never been reported to select wild-type mtDNA spontaneously[19]. Here too, both chemicals induced segregation to wild-type mtDNA (Fig. 1d); and they were most effective at decreasing m.3243A>G mutant load in primary patient-derived fibroblasts (P1 and P2) (Fig. 1e, f and Supplementary Fig. 1c). Further analysis of the patient-derived cells indicated that the increase in wild-type mtDNA was sufficient to restore mitochondrial translation, OXPHOS components and respiration to control levels (Fig. 1g–i and Supplementary Fig. 1d). The decrease in m.3243A>G was not accompanied by alteration of the mtDNA copy number (Supplementary Fig. 1e), nor did 2DG treatment cause cell death (Supplementary Fig. 2a, b). Segregation to wild-type mtDNA was also independent of cellular proliferation, as 2DG slowed cell growth equally in mutant and control cells (Supplementary Fig. 2c), and 2DG and 5TG decreased the mutant load in fully confluent (contact inhibited) P1 and P2 fibroblasts, again accompanied by restoration of OXPHOS protein levels (Fig. 1j and Supplementary Fig. 2d). Combined, these data indicate that the glucose analogues induce the selection of wild-type mtDNA in three different cell types (nuclear backgrounds) and on mtDNAs derived from three unrelated affected individuals, via intracellular selection, thereby achieving the goal of overriding the cells' propensity to maintain or select mutant mtDNAs.

The preceding experiments all employed 10 mM 2DG in medium containing 25 mM glucose and so, with an eye to future translational medicine studies, lower doses of 2DG were tested in combination with 5 mM glucose, a more physiological concentration. A dose of 0.5 mM 2DG proved sufficient to decrease the proportion of m.3243A>G (Fig. 2a), and 2DG was equally effective at lowering mutant loads when an escalating dose regime was applied (Fig. 2b).

The consistent effect of the 2DG and 5TG on the mutant load is clear evidence of them inducing active selection (Figs. 1 and 2a, b). To quantify the selective advantage to wild-type mtDNAs conferred by the glucose analogues, we calculated the heteroplasmy shift rate, using the entire time series dataset for each treatment (Fig. 2c). In the patient-derived fibroblasts, the glucose analogues produced a shift of −0.50, which corresponds to a change from 57% to 45% mutant mtDNA, in 1 week. Hence, our findings demonstrate that active selection can produce pronounced and rapid changes in heteroplasmy in human somatic cells.

### 2DG and 5TG restrict mtDNA replication in cells with high mutant load.
A decrease in the proportion of mutant mtDNA could be achieved by enhanced degradation of m.3243A>G or by selective inhibition of its replication. However, autophagic flux in m.3243A>G cells was repressed, rather than activated, by 2DG, suggesting that mtDNA turnover via autophagy is inhibited in these conditions (Fig. 3a and Supplementary Fig. 3a). Furthermore, there

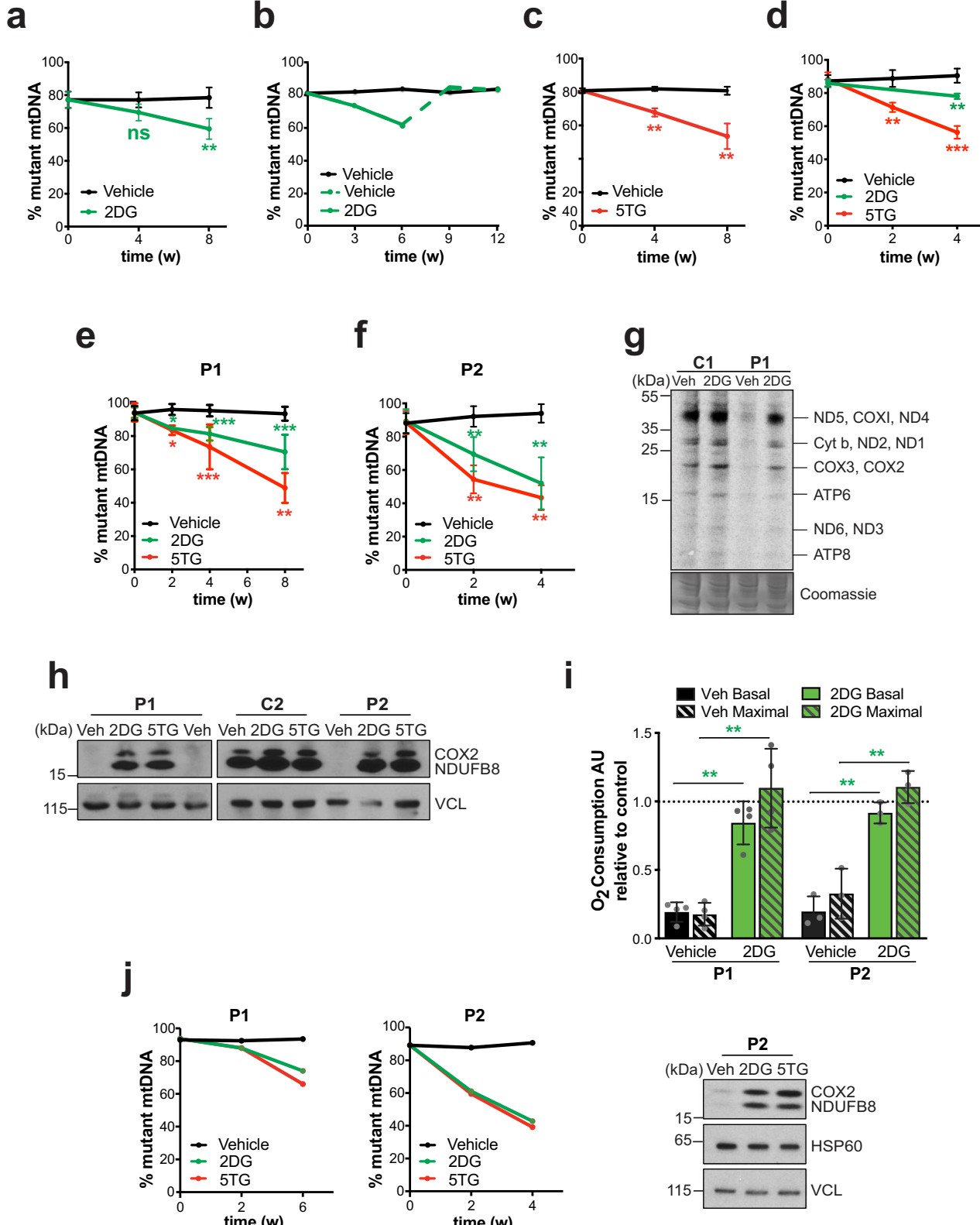

was no evidence that 2DG increased mtDNA turnover, based either on copy number measurements or on DNA immunostaining (Fig. 3b and Supplementary Figs. 1e and 3b). On the other hand, acute 2DG treatment markedly decreased bromodeoxyuridine (BrdU) pulse labelling of mtDNA in m.3243A>G cells, when compared to the same cells without 2DG or 2DG-treated control fibroblasts (Fig. 3b and Supplementary Fig. 3c). 5TG also repressed

mtDNA replication and autophagic flux in mutant cells, albeit to a lesser extent than 2DG (Fig. 3c and Supplementary Fig. 3d).

The marked inhibition of mtDNA synthesis in heteroplasmic m.3243A>G cells treated with the glucose analogues strongly suggested that the compounds specifically restrict the replication of the mutant mtDNA. Concordantly, de novo mtDNA synthesis increased by an order of magnitude after 4 weeks of 2DG

**Fig. 1 2-Deoxy-D-glucose (2DG) and 5-thioglucose (5TG) induce a shift from m.3243A>G to wild-type mtDNA on three nuclear backgrounds and restore mitochondrial respiratory capacity. a–f** The level of mutant (m.3243A>G) mtDNA was determined by pyrosequencing or restriction fragment length polymorphism analysis of DNA isolated from cells subjected to intermittent treatment with vehicle (black lines) or 10 mM 2DG (solid green line) or 10 mM, 5TG (red line). **a–c** A549 adenocarcinoma cells heteroplasmic for m.3243A>G. **b** After 42 days, A549 cells subject to continuous 10 mM 2DG treatment (solid green line) were grown for a further 42 days in medium lacking 2DG (broken green line). **d** Rhabdomyosarcoma (Myo.RD) m.3243A>G. **e, f** Primary human fibroblasts of patients (P1 and P2) carrying m.3243A>G. Number of independent experiments: $n = 5$ for (**a**); $n = 6$ for (**c**); $n = 7$ and 5 for 5TG and 2DG, respectively, panel (**d**); $n = 12$ and 9 for 2DG and 5TG, respectively, panel (**e**); $n = 8$ and 6 for 2DG and 5TG, respectively, panel (**f**); panels (**b**) and (**j**), $n = 1$. Data represent mean ± SD; two-sided Mann–Whitney test; ns, not significant; (**a**) ns $P$(A549 Veh vs. 2DG 4w) = 0.0952; (**a**) **$P$(A549 Veh vs. 2DG 8w) = 0.0079; (**c**) **$P$(A549 Veh vs. 5TG 4w) = 0.0079; (**c**) **$P$(A549 Veh vs. 5TG 8w) = 0.0022; (**d**) **$P$(Myo.RD Veh vs. 5TG 2w) = 0.0079; (**d**) ***$P$(Myo.RD Veh vs. 5TG 4w) = 0.0006; (**d**) **$P$(Myo.RD Veh vs. 2DG 4w) = 0.0079; (**e**) *$P$(P1 Veh vs. 5TG 2w) = 0.0286; ***$P$(P1 Veh vs. 5TG 4w) = 0.0006; **$P$(P1 Veh vs. 5TG 8w) = 0.0079; *$P$(P1 Veh vs. 2DG 2w) = 0.0286; ***$P$(P1 Veh vs. 2DG 4w) = 0.000041; ***$P$(P1 Veh vs. 2DG 8w) = 0.000011; (**f**) **$P$(P2 Veh vs. 2DG 2w) = 0.0022; (**f**) **$P$(P2 Veh vs. 2DG 4w) = 0.0022; (**f**) **$P$(P2 Veh vs. 5TG 2w) = 0.0022; (**f**) **$P$(P2 Veh vs. 5TG 4w) = 0.0079. Panels (**a**, **c–f**) with all the data points are showed in Supplementary Fig. 10. Increased wild-type mtDNA was accompanied by elevated mitochondrial translation in P1 vs. control (C1) (**g**); and increased OXPHOS proteins (**h**) and respiration (**i**) in P1 and P2. **g** Newly synthesized mitochondrial-encoded proteins in P1 fibroblasts were detected by $^{35}$S-methionine pulse labelling, $n = 3$ independent experiments; putative mitochondrial polypeptides labels are indicated to the side of the gel (NADH dehydrogenase subunits, ND1-6/L; cytochrome c oxidase subunits COX1-3; ATP synthase subunits 6 and 8; and cytochrome b); and Coomassie staining of the gel shows equal protein loading. **h** Steady-state levels of selected oxidative phosphorylation (OXPHOS) proteins (COX2 and NDUFB8) were detected by immunoblotting; vinculin (VCL) was used as loading control; $n = 11$ independent experiments. **i** Oxygen (O$_2$) consumption was measured by flux analysis (Seahorse Instrumentation); data in arbitrary units (AU) represent mean ± SD of $n = 4$ and 3 independent experiments for P1 and P2, respectively; two-sided unpaired $t$-test with Welch's correction: **$P$(P1 Veh vs. 2DG basal OCR) = 0.0014, **$P$(P1 Veh vs. 2DG maximal OCR) = 0.0055, **$P$(P2 Veh vs. 2DG basal OCR) = 0.0014, **$P$(P2 Veh vs. 2DG maximal OCR) = 0.0057. **g–i** Fibroblasts of P1 and P2 were treated intermittently with 2DG for 8 and 4 weeks, respectively. **j** Non-dividing P1 and P2 fibroblasts (by contact inhibition) were treated intermittently with the small molecules for 6 and 4 weeks, respectively. Immunoblots of P2 cellular protein for OXPHOS subunits are shown beside the chart showing the change in mutant load with time. Source data are provided as a Source Data file.

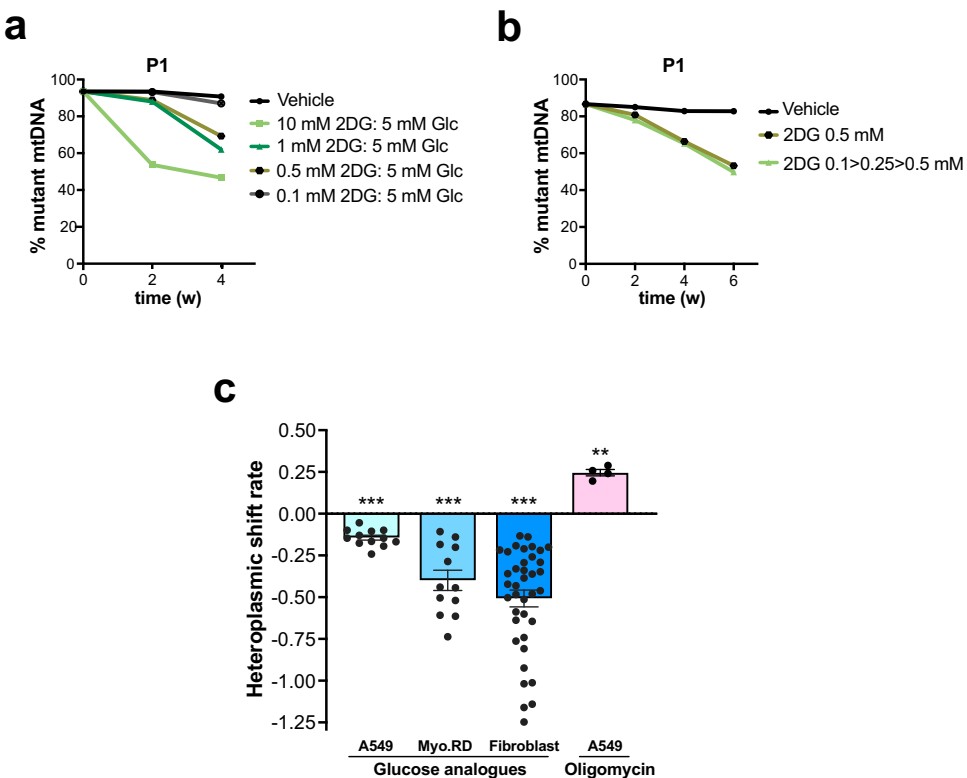

**Fig. 2 Minimal and escalating 2DG doses decreasing the m.3243A>G load, and overall shift rate. a** Fibroblasts were subjected to intermittent treatment with vehicle or different concentrations of 2DG (0.1–10 mM) with 5 mM glucose, or (**b**) an escalating dose regime, in which the first 2 rounds of treatment were 0.1 mM 2DG, followed by 2 rounds of 0.25 mM 2DG, and subsequent rounds of 0.5 mM 2DG. **c** Calculated heteroplasmy shift rates in A549 and Myo.RD cells, and primary fibroblasts, treated with and without glucose analogues (2DG or 5TG) or 1 mM glucose, no glutamine, or oligomycin. Shift rate describes the rate of change of heteroplasmy over time, accounting for the fact that it is measured as a percentage and hence follows sigmoidal dynamics. Number of independent experiments for 2DG and 5TG treatments: A549, $n = 12$; MyoRD, $n = 12$; fibroblasts, $n = 38$; for oligomycin A549, $n = 4$. Data represent mean ± SEM. $P$-values are given from a two-sided one-sample median test for A549 and Myo-RD, and a two-sided Wilcoxon signed-rank test for fibroblasts against the null hypothesis that glucose analogues have no effect on mutant load. $P$-values: A549 = 8.30E-07; Myo-RD: 3.90E-0.5; fibroblasts: 7.276E-12; A549 + Oligo: 0.001. Source data are provided as a Source Data file.

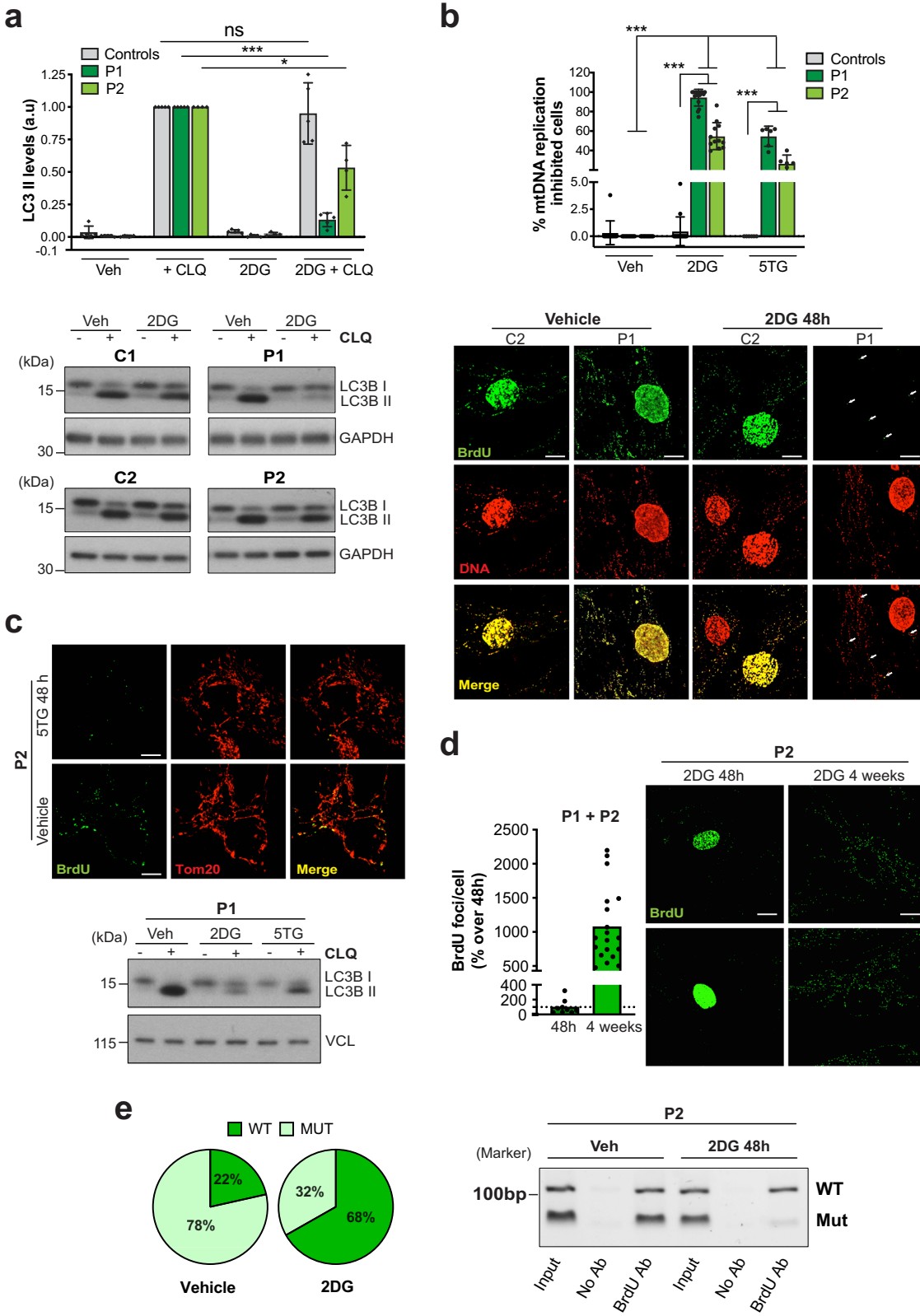

treatment in the m.3243A>G fibroblasts (Fig. 3d and Supplementary Fig. 3e), a period during which the wild-type mtDNA typically increased by a mean of 27 percentage points in the case of 2DG and 37 for 5TG (Fig. 1e, f and Supplementary Fig. 1c). As further confirmation, we performed a direct test, immunoprecipitating BrdU-labelled DNA from heteroplasmic cells treated with or without 2DG for 48 h, followed by analysis of

the mutant load. While in untreated cells, BrdU antibody captured wild-type molecules in a similar proportion to the total mtDNA (1.03:1), in 2DG-treated cells the wild-type mtDNA was enriched 3.25-fold by immunoprecipitation (Fig. 3e and Supplementary Fig. 3f). This result demonstrates that wild-type mtDNAs have a direct replicative advantage over mutants in the presence of 2DG.

**Fig. 3 2DG inhibits mtDNA replication and autophagy in fibroblasts with high mutant loads. a** Control (grey) and patient, P1 and P2, (green) fibroblasts treated with vehicle (veh) or 10 mM 2DG for 48 h for the final 6 h some cells were treated additionally with 50 μM choloroquine (CLQ) to block autophagy, and cellular proteins were analysed by immunoblotting. The chart indicates the ratio of the non-lipidated (I) and lipidated (II) autophagosome marker LC3, normalized to that of vehicle and CLQ-treated cells after normalization to the loading control. Data represent mean ± SD of n = 5 and 4 independent experiments for P1 and P2, respectively. Two-sided unpaired t-test with Welch's correction: ns $P$(Controls CLQ vs. 2DG + CLQ) = 0.6600; ***$P$(P1 CLQ vs. 2DG + CLQ) = 0.000003; *$P$(P2 CLQ vs. 2DG + CLQ) = 0.0122. **b** Control (C1-C3) and patient (P1 and P2) fibroblasts treated with and without 10 mM 2DG or 5TG for 48 h and labelled with 50 μM 5-bromo-2'-deoxyuridine (BrdU) for 13–16 h. After fixing, the cells were stained green with anti-BrdU and red with anti-DNA antibodies. Scale bar = 10 μm. Cells with little or no cytoplasmic BrdU-positive foci were scored as 'mtDNA replication inhibited' and ≥500 cells were counted from 6 independent experiments for 5TG, and >10 for 2DG (see also Supplementary Fig. 3c, d); data represent mean ± SD. One-way ANOVA: ***$P$(Controls vs. P1, 2DG; Controls vs. P2, 2DG; Controls vs. P1, 5TG; Controls vs. P2, 5TG; P1 Veh vs. 2DG; P1 Veh vs. 5TG; P2 Veh vs. 2DG; P2 Veh vs. 5TG) < 0.000001; ns $P$(Controls Veh vs. 2DG; Controls Veh vs. 5TG) > 0.9999. **c** m.3243A>G mutants treated with and without 5TG were labelled with BrdU and imaged as panel (**b**) for P2 fibroblasts (here, in red is stained the mitochondrial network with Tom20) or analysed for LC3 as panel (**a**) for P1 (n = 6 independent experiments, 3 each for P1 and P2). Scale bar = 10 μm. **d** BrdU labelling as panel (**b**), indicating the recovery of mtDNA synthesis in mutant (P1 and P2) cells after treatment with 2DG for 4 weeks compared to a 48 h treatment (n = 5 independent experiments, 3 for P2 and 2 for P1); data are expressed as BrdU foci (in the cytoplasm, i.e. newly synthesized mtDNA) per cell. Images are for P2 cells treated with 2DG for 48 h or intermittently for 4 weeks (see also Supplementary Fig. 2e for P1 and P2). Scale bar = 15 μm. **e** Patient (P2) fibroblasts were treated with 10 mM 2DG or vehicle. After DNA immunoprecipitation with an anti-BrdU antibody, a region of mtDNA encompassing 3243 bp was amplified and subjected to restriction fragment length polymorphism analysis. Pie charts represent the % of m.3243A>G (MUT - light green) and wild-type (WT - dark green) mtDNA in captured DNA fractions. Data represent mean ± SEM of n = 4 independent experiments. Two-sided unpaired t-test with Welch's correction: *$P$(Wild-type IP Veh vs. 2DG) = 0.0018. To the right, a representative agarose gel showing analysis of the anti-BrdU antibody precipitated material after PCR amplification and ApaI digestion, together with a mock-precipitation without antibody (No Ab), and the source DNAs prior to immunoprecipitation (Input). Without 2DG, the ratio of BrdU-containing wild-type mtDNA (IP fraction) to the total (Input) was 1.03:1 (n = 3 independent experiments). In contrast, with 2DG the ratio was 3.25:1 (n = 3 experiments); see Supplementary Fig. 3f for individual data points and chart. Hence, mtDNA synthesis (BrdU incorporation) is equal for mutant and wild-type molecules without treatment, whereas synthesis occurs predominantly on wild-type mtDNAs in the presence of 2DG. Source data are provided as a Source Data file.

**2DG and 5TG de-energize cells with mutant mtDNA.** Almost all mitochondrial activities depend on the maintenance of a proton gradient across the inner mitochondrial membrane. In the fly, it has been shown that dissipating the proton gradient arrests mtDNA replication, and mitochondria carrying a deleterious mtDNA variant are more readily de-energized under restrictive conditions, which limit mtDNA replication and permit the selection of wild-type molecules[13]. Therefore, we next determined the impact of 2DG and 5TG on the bioenergetics of cells with and without mutant mtDNA, via assays of ATP levels and mitochondrial depolarization. Without treatment, when mtDNA replication was not compromised, ATP levels were 80% of control values in patient-derived fibroblasts (Fig. 4a), despite respiration being markedly impaired (Fig. 1i, and[3,20]). Nor did inhibition of mitochondrial ATP production with oligomycin significantly alter ATP levels in either control or mutant cells. However, 2DG caused a much larger decrease in cellular ATP in m.3243A>G cells than controls (Fig. 4a). Combined, 2DG and oligomycin decreased the ATP level in all the cell lines much more than oligomycin alone (Fig. 4a); hence, 2DG targets an important source of non-mitochondrial ATP, i.e. glycolysis to lactate, especially in the m.3243A>G cells. Likewise, 2DG increased the proportion of depolarized mitochondria considerably more in mutant than in control cells, 25% and 5% of the total, respectively (Fig. 4b). The dearth of ATP in m.3243A>G cells treated with 2DG was further evidenced by activation of AMP kinase (AMPK), the cells chief energy sensor (Fig. 4c). Together, these data suggest that the compounds cause a severe energy shortage in cells containing m.3243A>G, which could inhibit the replication of the mutant mtDNA limiting its propagation, whereas the mitochondria with some wild-type mtDNA— and thus able to respire—continue to replicate mtDNA.

**2DG impairs mtDNA replication and autophagy when complex I is inhibited.** If, as we infer, mitochondrial fitness is important for mtDNA replication, then co-treatment of control cells with 2DG and the complex I inhibitor rotenone should mimic m.3243A>G cells treated with 2DG and inhibit mtDNA synthesis. Accordingly,

while rotenone alone had little effect on mtDNA synthesis, the two compounds together inhibited mtDNA synthesis in control cells, greater than, or equal to, 2DG in the respiratory-deficient m.3243A>G cells (Fig. 4d vs. Fig. 3b, d and Supplementary Fig. 3c, e). The 2DG/rotenone co-treatment of control cells was also equivalent to the 2DG treatment of m.3243A>G cells with respect to inhibition of autophagic flux and AMPK activation (Fig. 4e vs. Fig. 3a and Supplementary Fig. 3a). These data indicate that 2DG forces control cells to depend on mitochondrial energy production for mtDNA replication and autophagy. The findings can also explain how 2DG has a greater impact on mitochondria with m.3243A>G than those with wild-type mtDNA in heteroplasmic cells: the mutant mitochondria are complex I deficient and so equivalent to mitochondria of control cells treated with 2DG and rotenone, whereas replication should remain active in the few mitochondria with wild-type mtDNA, as they possess a functional respiratory chain.

Thus, the model emerging is that the wild-type mtDNA derives its selective advantage over m.3243A>G from the fact that replication becomes respiration/complex I dependent in the presence of 2DG. If true, a combined rotenone and 2DG treatment should negate any selective advantage of wild-type mtDNA conferred by 2DG in heteroplasmic m.3243A>G cells, as should other inhibitors of mitochondrial energy production. The primary fibroblasts carrying m.3243A>G did not survive long-term treatments with OXPHOS inhibitors; however, in A549 cells with m.3243A>G, rotenone with 2DG reversed the direction of mtDNA segregation, compared to 2DG alone (Fig. 4f, g); and oligomycin-induced inhibition of ATP synthase also promoted segregation to mutant mtDNA on the A549 background (Fig. 4h). Negation of mitochondrial fitness by the OXPHOS inhibitors will permit the selection of the mutant mtDNA by selfish mechanisms, such as decreased replication pausing[21].

**Cells with mutant mtDNA are more reliant on glutamine for mtDNA replication than controls.** Although glucose metabolism is the most obvious target of 2DG to affect cellular bioenergetics and mitochondrial fitness, the compound also inhibits glutamine

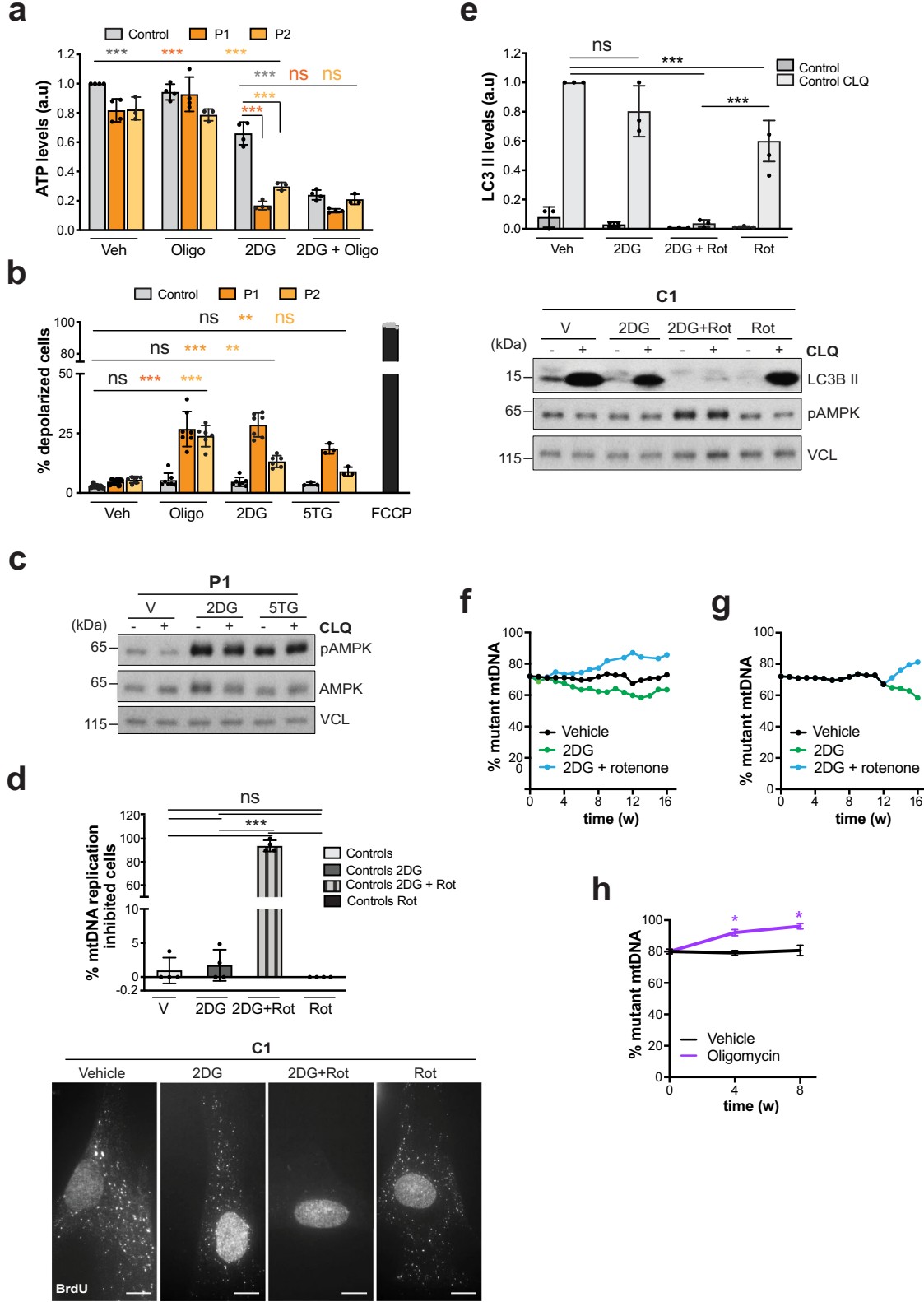

utilization;[18] a process that could provide critical support to the replication of mutant mtDNA, given that cells with mitochondrial dysfunction are heavily reliant on glutamine[16]. Therefore, we assessed the contributions of glucose and glutamine to mtDNA replication by restricting their availability, adding rotenone in some experiments as a 'm.3243A>G mimetic'. Glutamine withdrawal inhibited mtDNA replication in the cells with a high

mutant load, much more than in control cells (Fig. 5, panels 1 vs. 3 (control) and 5 vs. 7 (m.3243A>G)), demonstrating that the increased glutamine consumption associated with mitochondrial dysfunction[16] is supporting mtDNA replication. Moreover, as neither glucose restriction (Fig. 5, panel 13), nor replacement of glucose with galactose (prior to cell death) (Supplementary Fig. 5), inhibited mtDNA synthesis in mutant cells, we conclude

**Fig. 4 Bioenergetics underlies the effects of 2DG and 5TG on mtDNA replication and autophagy in fibroblasts with high mutant load, and inhibition of OXPHOS reverses the direction of segregation. a** ATP levels in Control (grey), P1 (orange) and P2 (light orange) fibroblasts treated without and with 10 mM 2DG for 48 h and 1 μM oligomycin (Oligo) for 90 min, or both compounds. Data represent mean ± SD of $n = 4$ and 3 independent experiments for P1 and P2, respectively. One-way ANOVA: ***$P$(Control Veh vs. 2DG; P1 Veh vs. 2DG; P2 Veh vs. 2DG) < 0.000001; ***$P$(Control vs. P1, 2DG; Control vs. P2, 2DG) < 0.000001; ***$P$(Control 2DG vs. 2DG + Oligo) < 0.000001; ns $P$(P1 2DG vs. 2DG + Oligo) = 0.5867; ns $P$(P2 2DG vs. 2DG + Oligo) = 0.1885. **b** Proportion of depolarized cells in Control, P1 and P2 fibroblasts treated without and with 2DG, 5TG or Oligo normalized to 5 μM FCCP-treated cells (100% depolarized cell). Data represent mean ± SD of $n = 7$ and 6 independent experiments for P1 and P2 cells treated with 2DG, respectively; and $n = 3$ and 3 for 5TG. One-way ANOVA: ns $P$(Control Veh vs. Oligo) = 0.0942; ns $P$(Control Veh vs. 2DG) = 0.2868; ns $P$(Control Veh vs. 5TG) = 0.886; ***$P$(P1 Veh vs. Oligo) < 0.000001; ***$P$(P1 Veh vs. 2DG) < 0.000001; **$P$(P1 Veh vs. 5TG) = 0.0032; ***$P$(P2 Veh vs. Oligo) < 0.000001; ***$P$(P2 Veh vs. 2DG) = 0.0009; ns $P$(P2 Veh vs. 5TG) = 0.2631. **c** P1 fibroblasts were treated with and without 2DG, 5TG and CLQ as indicated. Extracted cellular protein was immunoblotted for AMP kinase (AMPK), and the activated form phosphorylated at threonine 172 (pAMPK) and vinculin (VCL) as the loading control; $n = 4$ independent experiments. **d** Mitochondrial DNA synthesis in control fibroblasts treated with (mid-grey) and without (light grey) 10 mM 2DG, 1 μM rotenone (black), or 10 mM 2DG and 1 μM rotenone (striped mid-grey) for 48 h; BrdU labelling and analysis as Fig. 3b. Data represent mean ± SD of $n = 4$ independent experiments. One-way ANOVA: ns $P$(Controls Veh vs. 2DG) = 0.9787; ***$P$(Controls Veh vs. 2DG + Rot) < 0.000001; ns $P$(Controls Veh vs. Rot) = 0.9628; ***$P$(Controls 2DG vs. 2DG + Rot) < 0.000001; ns $P$(Controls 2DG vs. Rot) = 0.8211; ***$P$(Controls 2DG + Rot vs. Controls Rot) < 0.000001. Scale bar = 10 μm. **e** Autophagic flux in control cells treated with and without 10 mM 2DG and 1 μM rotenone (rot) for 48 h; immunoblot and data analysis as Fig. 3a. Controls without (mid-grey) and with (light grey) CLQ. Data represent mean ± SD of $n = 3$ independent experiments. One-way ANOVA: ns $P$(Control CLQ Veh vs. CLQ + 2DG) = 0.1403; ***$P$(Control CLQ Veh vs. CLQ + 2DG + Rot; Control CLQ + 2DG + Rot vs. CLQ + Rot) < 0.000001; ***$P$(Control CLQ Veh vs. CLQ + Rot) = 0.0005. A549 cells were treated with vehicle (black), 10 mM 2DG (blue) or 2DG and 1 μM rotenone (green) (**f, g**), or oligomycin (purple) alone (**h**), and m.3243A>G mutant load measured at intervals by pyrosequencing. In panel (**g**) the vehicle-treated cells were split at day 84 and a new 2DG or 2DG + rotenone treatment commenced (vehicle-treated values beyond day 84 are shown in panel (**f**), but not in (**g**)). **h** Data represent mean ± SD of $n = 4$ independent experiments. Two-sided, Mann-Whitney: *$P$(A549 Veh vs. Oligomycin 4w; A549 Veh vs. Oligomycin 8w) = 0.0286. Source data are provided as a Source Data file.

---

that restricting glutamine utilization is the means by which 2DG inhibits the replication of mutant mtDNA.

Notwithstanding the above, further analyses indicated that glucose is critical for mtDNA replication in some contexts, and that rotenone does not model all the features of m.3243A>G. While the 2DG/rotenone co-treatment inhibited mtDNA replication in controls similar to 2DG in mutant cells (Fig. 4d), glutamine withdrawal and rotenone treatment in controls (Fig. 5, panel 4) did not mimic the effect of glutamine deprivation observed in mutants (Fig. 5, panel 7). Instead, it was necessary to restrict glucose (to 1 mM), as well as to add rotenone and withdraw glutamine, to block replication in control cells (Fig. 5, panel 12). Evidently, the 1 mM glucose is utilized by Complex I to support mtDNA replication, given that mtDNA synthesis is active when both glucose and glutamine are restricted unless respiration is inhibited with rotenone (Fig. 5, panel 11 vs. 12). Nevertheless, glycolysis and complex I are dispensable for mtDNA synthesis. Not only was replication maintained upon rotenone treatment in 1 mM glucose in both control and mutant cells (Fig. 5, panel 10), it was also active when the cells were treated with the glycolytic (GAPDH) inhibitor koningic acid (KA)[22,23], with and without rotenone (Supplementary Fig. 4).

This web of comparisons distils down to the conclusion that cells have three means of sustaining mtDNA replication: (1) glycolysis; (2) glucose-supported respiration via complex I; and (3) glutamine metabolism. Their respective contributions are not fixed but vary according to metabolite availability and pathway flux; i.e. they are in a state of dynamic equilibrium. It then becomes evident that complex I deficiency, by increasing glycolytic ATP production and elevating glutamine consumption in the mutant cells, will make the replication of their mtDNA susceptible to changes in glucose and glutamine distinct from control cells. These distinctions, particularly as regards complex I activity, will apply equally at the intracellular level, where there is a mixture of mitochondria with mutant and wild-type mtDNA in the heteroplasmic state. In conclusion, mtDNA replication is dependent on nutrient availability and mitochondrial (dys)function'.

**Combined glutamine and glucose restriction promotes the selection of wild-type mtDNA equal to 2DG.** Because 2DG

limits the utilization of glutamine and glucose, we next investigated their effects on the mtDNA selection directly. We maintained m.3243A>G heteroplasmic cells in our standard medium containing 25 mM glucose and 4 mM glutamine; the same medium but without glutamine; or media with 1 mM glucose with and without glutamine. Mutant cells in 1 mM glucose with glutamine grew more slowly than control cells (Supplementary Fig. 6), which would drive selection via intercellular competition; additionally, they started dying in substantial numbers after three pulses of this medium. Thus, the specific effect of glucose restriction on mtDNA segregation could not be assessed. In contrast, the combined glutamine and glucose restriction regime slowed growth equally in control and mutant cells (Supplementary Fig. 6), as per 2DG (Supplementary Figs. 2c and 6), and was not associated with cell death, indicating that the absence of glutamine enhances mutant cell survival when glucose is limiting. While glutamine withdrawal produced a modest increase in the level of wild-type mtDNA, it was the combination of glutamine and glucose restriction that recapitulated the effect of 2DG on segregation (Fig. 6a), leading to increases in OXPHOS protein levels and mtDNA synthesis (Fig. 6b, c). Taken together, the data suggest that while mutants are more dependent on glutamine, functional wild-type mtDNA are more reliant on complex I-dependent energy production (when glucose is scarce) and can use this to maintain the mitochondrial membrane potential. Therefore, we infer that combined glutamine and glucose restriction imposes twin selective pressures—negative on the mutant and positive on wild-type mtDNAs—and that 2DG is effective at driving segregation to wild-type molecules because it restricts the utilization of both substrates.

In addition to inhibiting glycolysis and restricting glutamine utilization[17,18], 2DG induces ER stress[24,25], as it is structurally similar to mannose. We confirmed that 2DG increased GRP78 expression and that this was attenuated by mannose, without inactivating AMP kinase (Supplementary Fig. 7a, b). Nevertheless, 2DG with mannose was at least as effective at inducing segregation to wild-type mtDNA as 2DG alone (Supplementary Fig. 7c, d). Therefore, 2DG's effect on mtDNA segregation does not relate to its similarity to mannose, nor GRP78-related ER stress.

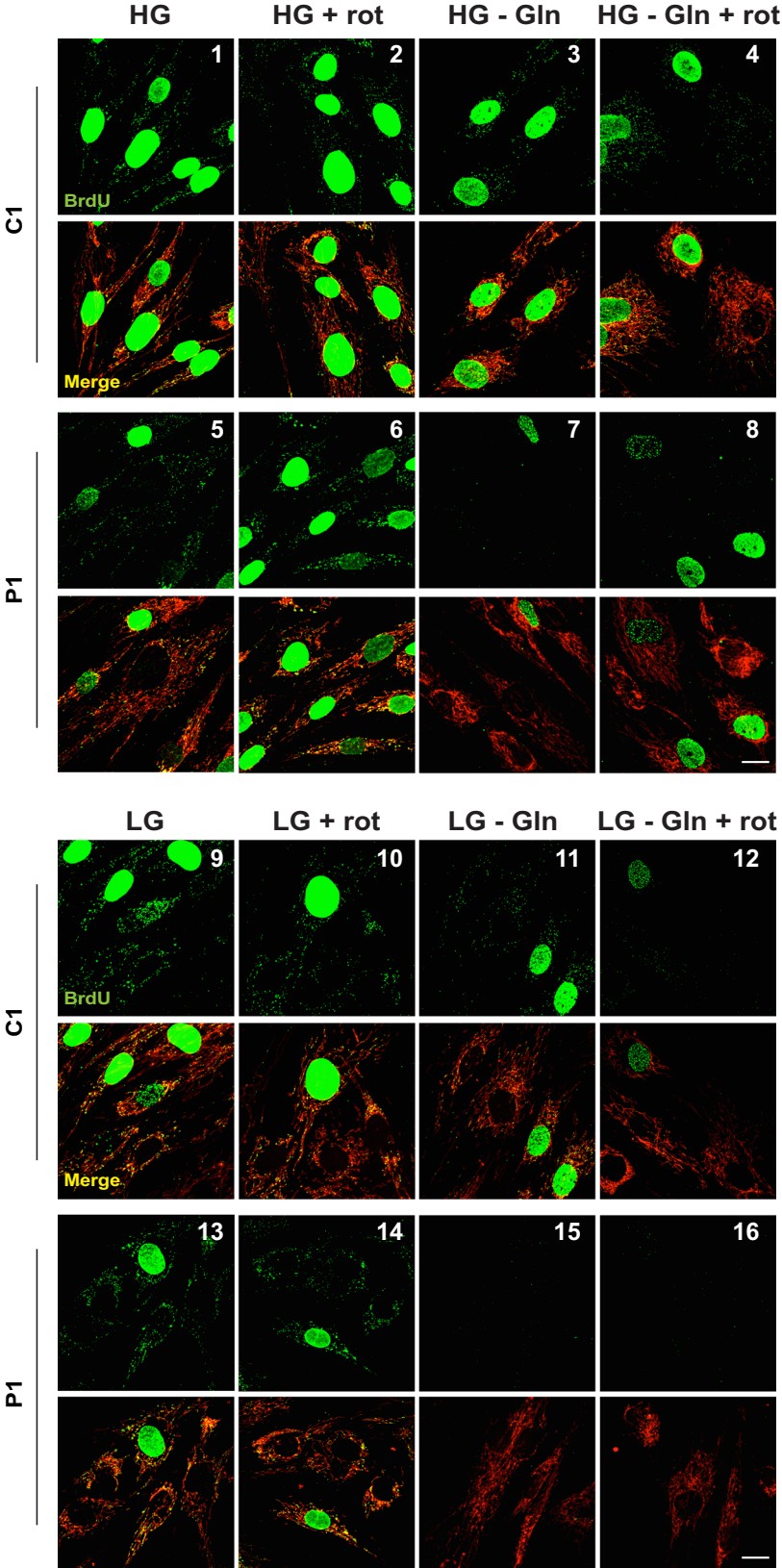

**Fig. 5 Glutamine restriction preferentially inhibits mtDNA replication in m.3243A>G fibroblasts.** Control (C1) and patient (P1) fibroblasts were grown in medium containing or lacking glutamine and different concentrations of glucose for 24 h, with 50 μM BrdU to label newly synthesized mtDNA for the final 13 h; in some cases 0.5 μM rotenone was added together with the BrdU to assess the additional impact of inhibition of complex I. Cells were fixed and immunostained with BrdU (green) and Tom20 (red) to detect DNA that had incorporated BrdU in the mitochondrial network (merge). HG, 25 mM glucose; LG, 1 mM glucose; rot, rotenone; –Gln, no glutamine. Scale bar = 10 μm; n = 4 independent experiments.

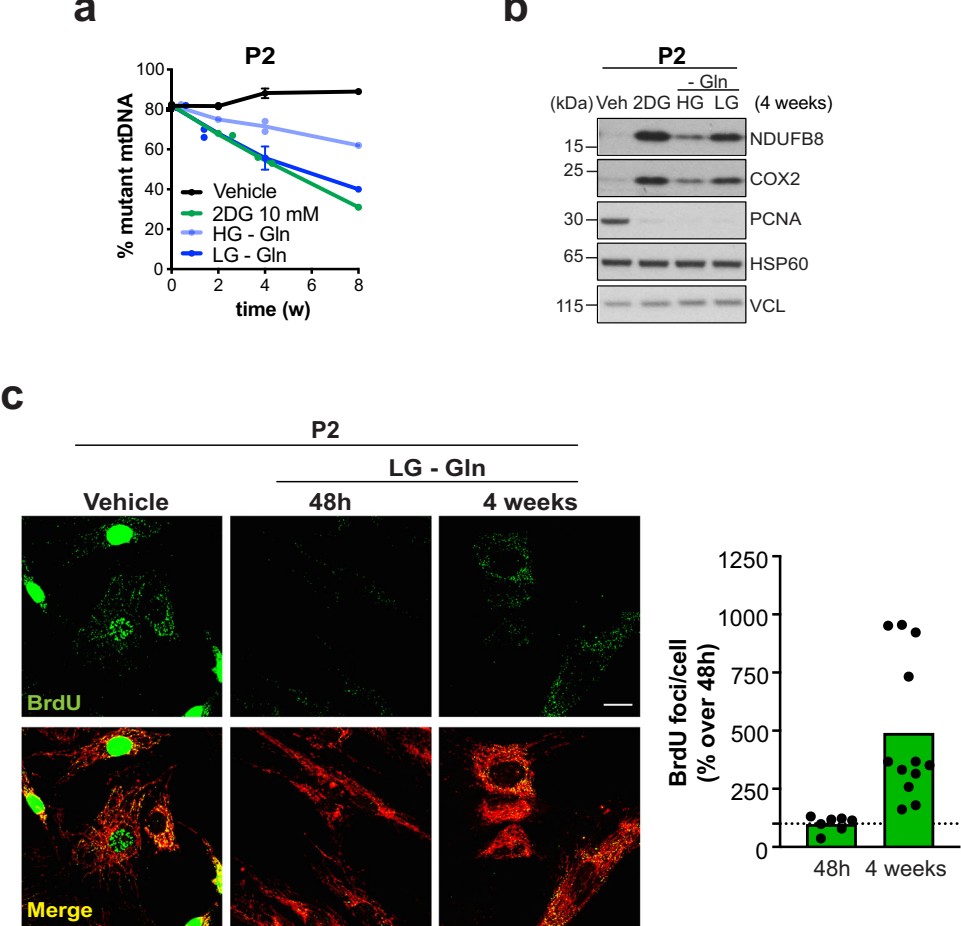

**Fig. 6 Glutamine restriction induces a shift from m.3243A>G to wild-type mtDNA in mutant fibroblasts, which is more pronounced in low-glucose conditions, mimicking the effect of 2DG. a** P2 fibroblasts were treated intermittently with vehicle (black line) or 10 mM 2DG (green line) or 25 mM glucose no glutamine (HG –Gln, light blue line) or 1 mM Glucose no glutamine (LG –Gln, dark blue line). The proportion of mutant mtDNA was plotted against time. Data represent mean ± SD of $n = 2$ experiments for 2DG and HG –Gln, and $n = 3$ for LG –Gln. For all the conditions one experiment was performed on contact inhibited cells. **b** Steady-state levels of the nuclear (NDUFB8) and mitochondrial (COX2) OXPHOS subunits after 4 weeks of intermittent treatment detected by immunoblotting. Proliferating cell nuclear antigen (PCNA) was used as an indicator of cell proliferation, HSP60 as an indicator of mitochondrial mass, and VCL as the loading control; $n = 2$ independent experiments. **c** m.3243A>G mutants (P2) treated with vehicle or LG –Gln for 48 h or 4 weeks were labelled with BrdU and imaged as previously (Fig. 3b). The BrdU signals indicate the recovery of the mtDNA synthesis after 4 weeks vs. 48 h. $n = 2$ independent experiments; scale bar = 15 μM. In the accompanying chart the data are expressed as cytoplasmic BrdU foci (i.e. replicating mtDNAs) per cell (green columns). Source data are provided as a Source Data file.

## Discussion

Pathological variants of human mtDNA were first reported over 30 years ago[26–28], and since then the primary goal in the field has been to identify ways of selecting against mutant variants with the aim of treating heteroplasmic mtDNA disorders. This study has identified in 2DG, and the related 5TG, small molecules that can purge cells of the common pathological mtDNA variant m.3243A>G, restoring mitochondrial respiratory capacity. Our analysis of the mechanism of action indicates that 2DG preferentially depolarizes mutant mitochondria, inhibits the replication of mutant mtDNA and allows the propagation of functional mtDNA molecules. A temperature-sensitive mtDNA variant in flies behaved similarly[13], and so the current study indicates that this general mechanism of restricting the propagation of deleterious variant operates both in somatic and germline cells. Moreover, as we show that restricting glutamine and glucose recapitulates the effects of 2DG on mtDNA replication and segregation, we conclude that both glucose and glutamine metabolism are critical targets of 2DG (Fig. 7). This opens up a new area of research—the regulation and control of mtDNA replication and propagation via the manipulation of nutrient metabolism.

Logically, the positive selection of wild-type mtDNA depends ultimately on its capacity to produce functional products for OXPHOS. Given that mtDNA replication is dependent on respiration via complex I when control cells are exposed to 2DG (Fig. 4d), or glutamine and glucose are restricted (Fig. 5, panel 11), it is respiration via complex I that gives the mitochondria with wild-type mtDNA their selective advantage. Contrarily, inhibition of OXPHOS reverses segregation to wild-type mtDNA (Fig. 4f–h), again indicating that mitochondrial fitness is essential for the maintenance and selection of functional molecules. That said, the demonstration that cells/mitochondria with high levels of m.3243A>G are replication-competent when glycolysis is reduced (Fig. 5, panel 13 and Supplementary Figs. 4 and 5) potentially explains how deleterious mtDNA variants of this type are transmitted in the human germline: mitochondria with m.3243A>G need not depend on glycolysis for mtDNA maintenance, and so the desired selective pressure might be lacking, if, for example, glutamine is readily available.

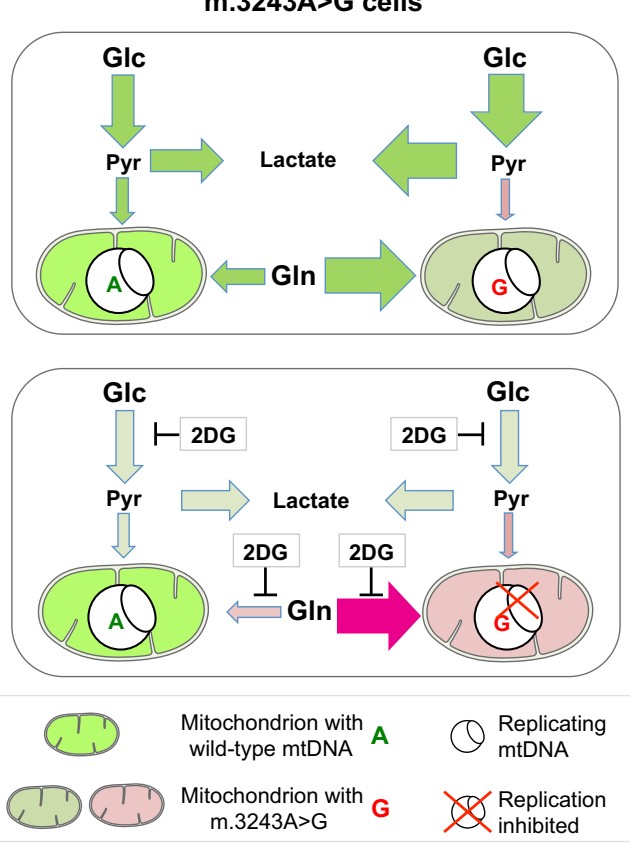

**m.3243A>G cells**

**Fig. 7 2DG promotes segregation to wild-type mtDNA by restricting glutamine (Gln) and glucose (Glc) utilization.** Mitochondrial DNA replication can be supported by glucose-fuelled respiration, glycolysis or glutamine (see text for details). The mitochondrial dysfunction caused by m.3243A>G disables the first of these, and consequently increases glycolysis and glutamine consumption[16]. 2DG restricts glucose and glutamine metabolism[17,18]. Hence, 2DG forces cells/mitochondria to rely on pyruvate for mitochondrial energy production to sustain mtDNA replication (Fig. 4d). This provides a model that explains how 2DG drives the positive selection of organelles with wild-type mtDNA (Fig. 2c), as mitochondria with m.3243A>G are respiratory (complex I)-deficient and so largely unable to utilize pyruvate. Bright green-filled arrows/mitochondria - highly active; light green-filled arrows/mitochondria - low activity; pink-filled arrows/mitochondria - impaired activity.

The identification of small molecules that promote the selection of wild-type mtDNA over mutant m.3243A>G—restoring mitochondrial protein synthesis and respiratory capacity—advances the prospect of pharmacological treatments for heteroplasmic mtDNA disorders. While the compounds have not been tested in patients with m.3243A>G and there is no direct equivalent of m.3243A>G in any animal, there is no doubt that 2DG can alter nutrient utilization in vivo, having been used safely in animal models of human diseases[29–31] and in humans[32], and is currently being tested in preclinical studies for epilepsy (https://ncats.nih.gov/bridgs/projects/active/2dg-treatment-epilepsy), making its potential translation to therapy more within reach. Moreover, increased glutamate utilization is a feature of mitochondrial dysfunction in a mouse model[16], and the predominant complex I deficiency associated with m.3243A>G mtDNA in cells[5,20] is also seen in muscle of subjects with this mutant[33]. Thus, the identified mechanism of action of 2DG is expected to be applicable in vivo; that is, the mutant mitochondria in vivo will also be more reliant on glutamine utilization than their wild-type

counterparts, while the latter benefit from much higher complex I capacity. Furthermore, the imposition of mitochondrial fitness selects against defective mtDNAs in flies[13], and in the mammalian germline the selection of functional mitochondria coincides with a switch from glycolytic to oxidative metabolism[14].

Although 5TG appears to be as effective as 2DG at positively selecting wild-type mtDNA, and it does not induce the pronounced ER stress of 2DG, it has never been tested in humans. Mannose, on the other hand, is an approved dietary supplement, which depresses the ER stress caused by 2DG (Supplementary Fig. 7a, b) without interfering in the selection of wild-type mtDNAs (Supplementary Fig. 7c). Therefore, the combination of 2DG and mannose may well be the best prospect for treatment of m.3243A>G and other heteroplasmic mtDNA disorders, and might offer a more tolerable treatment for epilepsy than 2DG alone.

## Methods

**Cell culture.** A549 adenocarcinoma and MyoRD rhabdiomyosarcoma m.3243A>G cybrid cells[5,11] were maintained in Dulbecco's Modified Eagle's Medium (DMEM) containing 25 mM glucose (Life Technologies) supplemented with 10% fetal bovine serum (FBS, Pan Biotech UK), 1 mM of pyruvate, 1% penicillin and streptomycin (PS, Life Technologies), at 37 °C in a 5% $CO_2$ atmosphere. Primary skin fibroblasts were grown in DMEM GlutaMAX$^{TM}$ (Life Technologies) with the same supplements. All the cell lines were regularly confirmed free of mycoplasma, using the Look Out Mycoplasma PCR Detection Kit (Sigma).

Glucose restriction employed glucose-free DMEM medium (Life Technologies) with the addition of no or 1 mM glucose, as indicated, whereas galactose was added to 5 mM, plus 10% dialyzed or non-dialyzed serum, also as indicated. For the glutamine restriction experiments, 1 mM or 25 mM glucose was added to DMEM lacking glutamine, supplemented with 10% dialyzed serum.

For the acute treatments, cells carrying m.3243A>G were grown to 50–60% confluent and treated for 24 or 48 h with the compounds and concentrations indicated in the main text, figures and methods. Intermittent treatments extending over several weeks comprised 48 or 72 h pulses with the drug or modified medium, followed by 24 or 48 h of recovery, throughout the course of the experiments (see Supplementary Fig. 8).

**Inhibitors and chemicals.** Chemicals were purchased from Sigma, except for koningic acid (Abcam). Rotenone, oligomycin and 5-bromo-2′-deoxyuridine were dissolved in DMSO, whereas all the other chemicals were dissolved in milliQ-grade water. The final concentrations were as follows: 10 mM 2-Deoxy-D-glucose (unless otherwise specified), 10 mM 5-thioglucose, 10 mM mannose, 0.5 or 1 μM rotenone, 50 μM chloroquine, 50 μM 5-bromo-2′-deoxyuridine, 1 μM oligomycin and 0.5 or 1 μM koningic acid.

**Cellular proliferation rate and vitality.** The cellular proliferation rate was determined using an IncuCyte Zoom cell imager (Essen Bioscience). In all, $3 \times 10^4$ cells were seeded in 6-well plates and imaged every hour for 3 days. The proliferation rate was determined using the Incucyte Zoom software 2015A. At the end of the treatment, the cells were labelled with 5 μM calcein (Molecular Probes, ThermoFisher Scientific) for 20 min and then imaged.

**Cytotoxicity assay.** The release of the lactate dehydrogenase (LDH) in the medium was measured adapting the instructions from the manufacturer (Cytotoxicity Detection Kit, Roche). Briefly, control and patient fibroblasts were seeded on multi-6-well plates (ThermoFisher Scientific) and subjected to either vehicle or 2DG treatment for 48 h or cell grown in galactose medium. Cells were seeded at different densities, taking in account the differences in their growth rate: $3 \times 10^4$ for vehicles and $6 \times 10^4$ for 2DG-treated cells. For the positive control, cells were treated with 1% Triton X-100 (Santa Cruz Biotechnology). Then, 100 μl of the medium was used for each assay. After incubating the medium with the dye for 30 min at room temperature, the absorbance at 490 nm was measured using a plate reader (Biorad). The data were then normalized for the protein content after cell lysis.

**Fractionation and immuno-detection of proteins.** Cells were lysed on ice with RIPA buffer (65 mM Tris, 150 mM NaCl, 1% Nonidet P-40), 0.25% Na-DOC, 1 mM EDTA, pH 7.4) 1x protease inhibitor cocktail (PIC, Roche), phosphatase inhibitor cocktail (Cell Signaling), 50 U Benzonase (Millipore). After incubating on ice for 20 min, the samples were centrifuged for 20 min at 13,000g, to separate the proteins from the DNA. Protein concentration was determined by DC protein assay kit (Biorad). Protein samples were prepared in 1x Laemmli loading buffer and resolved on 4–12%, 10% or 12% Bis-Tris NuPAGE gels (Life Technologies, ThermoFisher Scientific) run in NuPAGE MES or MOPS buffers (Life

Technologies, ThermoFisher Scientific). After electrophoresis, proteins were transferred to a polyvinylidene fluoride membrane (PVDF, Millipore) and blocked in 5% milk (Sigma), phosphate-buffered saline (PBS) containing 0.1% Tween (ThermoFisher Scientific) for 1 h. Membranes were incubated overnight with primary antibodies (see below), at 4 °C and, after washes, with the appropriate secondary antibodies for 1 h at room temperature. Proteins were detected using standard ECL$^{TM}$ Western Blotting Analysis System (GE Healthcare) or Super-Signal® West Dura (Thermo Scientific). Western blots were digitalized using a Canoscan 9000F scanner (Canon). Optical density quantification of bands detected by western blotting was carried out using the designated tools available with Fiji ImageJ (2.0.0-rc-15/1.49 h).

**Antibodies**. The following primary antibodies were used in this study: BrdU (Biorad, MCA2060 or Abcam, ab6326, 1:200 dilution); DNA (Progen, AC-30-10, 1:250 dilution); GAPDH (Sigma, G8795 or Abcam, ab8245, 1:10,000 and 1:2000 dilutions, respectively); GRP78 (Santa Cruz Biotech, Sc-13968, 1:1000 dilution); HSP60 (Abcam, ab46798, 1:1000 dilution); LC3B (Sigma, L7543) 1:5000; MTCO-2 (Abcam, ab110258 1:1000 dilution); NDUFB8 (Abcam, ab110242, 1:1000 dilution); AMPK alpha (Cell Signaling, 2532, 1:1000 dilution); phosphoAMPK alpha (Cell Signaling, 2531, 1:1000 dilution); TOM20 (Santa Cruz Biotech or Abcam, Ab186735, 1:4000 and 1:10,000 dilutions, respectively); VCL (Abcam, ab18058, 1:1000 dilution); and proliferating cell nuclear antigen (PCNA) (Mouse, sc-56, 1:8000 dilution).

Secondary Antibodies: Anti-Mouse IgG (H + L), HRP Conjugate (Promega, W4021, 1:4000 dilution); anti-Rabbit IgG (H + L), HRP Conjugate (Promega, W4011 1:4000 dilution); Alexa Fluor®−488 goat- anti-mouse (Invitrogen, A-10684,1:1000 dilution); Alexa Fluor®−568 goat- anti-mouse (Invitrogen, A-11004, 1:1000 dilution); Alexa Fluor® 568 donkey anti-rabbit (Invitrogen, A-10042, 1:1000 dilution); Alexa Fluor®-488 goat- anti-rat (Invitrogen, A-11006, 1:1000 dilution).

**Determination of mutant load**. DNA was extracted from cells using the Puregene system (Qiagen) or Wizard SV Genomic DNA Purification System (Promega), and the proportion of wild-type mtDNA and m.3243A>G was determined by pyrosequencing, which has been validated for quantification of m.3243A>G heteroplasmy[34]. Briefly, a 155 base pair region of human mtDNA encompassing the m.3243A>G site was amplified using the PyroMark PCR kit (Qiagen) (all primers are listed in Table S1). Pyrosequencing reactions were performed using a sequencing primer and PyroMark reagents (Qiagen) on a PSQ 96MA pyrosequencer and analysed with PSQ 96MA 2.1 software. Pyrosequencing exhibited a standard deviation range of 0.06–4.64% change in heteroplasmy across 359 samples measured in triplicate. Last-cycle PCR of sequence spanning 1155–1725 bp of human mtDNA that includes an invariant ApaI site was used as a positive control to confirm complete digestion. Alternatively, heteroplasmy was measured by restriction fragment length polymorphism analysis, using amplified mtDNA spanning 2966–3572 bp; and the mutant load was estimated from the proportion of DNA cleaved by ApaI, after separation of digested PCR product via agarose gel electrophoresis[12].

**BrdU-DNA immunoprecipitation**. Primary skin fibroblasts treated acutely with 10 mM 2DG received a 16 h pulse of 50 µM 5-bromo-2'-deoxyuridine (BrdU, Sigma). Parallel conditions without BrdU and/or without 2DG were used as controls. One microgram of isolated DNA was digested with ApoI restriction enzyme to generate ~300 bp fragments containing the m.3243A>G site. Fragmented DNA was then denatured for 10 min at 95 °C in PBS containing 10 µg of sheared Salmon Sperm DNA (Invitrogen) in a final volume of 50 µl. Samples were precleared with 50 µl of 50% Protein G Agarose beads solution (Thermo Scientific) for 2 h at 4 °C with constant shaking. Beads were pelleted by centrifugation and the supernatant incubated overnight at 4 °C with 1 µg of anti-BrdU primary antibody in 200 µl of PBS containing 0.625% triton X-100. Antibody–DNA complexes were captured with 50 µl of 50% Protein G Agarose beads solution (Thermo Scientific) for 1 h at 4 °C. Beads were then washed 3 × 5 min with 1% Triton X-100, 0.1% SDS, 150 mM NaCl and 2 mM EDTA in 20 mM Tris pH 8.0 followed by a final wash with 1% Triton-X100, 0.1% SDS, 500 mM NaCl, 2 mM EDTA in 20 mM Tris pH 8.0. DNA was eluted with 1% SDS in TE buffer for 15 min at 65 °C. For each sample, a second tube without antibody incubation was run in parallel as control. Eluates form each sample were purified with phenol:chloroform and resuspended in 20 µl TE buffer. The mutant load in the immunoprecipitated DNA was estimated by restriction fragment length polymorphism analysis of amplified mtDNA spanning 3202–3328 bp (see section 'Determination of mutant load').

**Quantification of the mtDNA copy number**. The mtDNA copy number was quantified as follows: after DNA isolation, real-time quantitative PCR was performed in triplicates on 384-Well Reaction Plates (Applied Biosystems) in final volumes of 10 µl. Each reaction contained 20 ng of DNA template, 1x Power SYBR-Green PCR Master Mix (Applied Biosystems) and 0.5 µM of forward and reverse primers. Mitochondrial and nuclear DNA were amplified using primers specific to regions of human COX2 and APP1 genes. Changes in the mtDNA copy number were determined by using the $2^{-\Delta\Delta Ct}$ method[35] and represented as fold-change relative to the mean value for vehicle-treated cells analysed in parallel[36].

**Mitochondrial translation assay**. Mitochondrial translation products were labelled using $^{35}$S-methionine[37]. Fibroblasts were washed twice with methionine/cysteine-free DMEM (Life Technologies) supplemented with 1 mM L-glutamax, 96 µg/ml cysteine (Sigma), 1 mM pyruvate and 5% (v/v) dialyzed FBS, and incubated in the same medium for 10 min at 37 °C. Then, 100 µg/ml emetine dihydrochloride (Sigma) was added to inhibit cytosolic translation, before pulse labelling with 100 µCi [$^{35}$S]-methionine for 45–60 min. Cells were chased for 10 min at 37 °C in regular DMEM with 10% FBS, washed three times with PBS and harvested. Labelled cells were lysed in PBS, 0.1% n-dodecyl-D-maltoside (DDM), 1% SDS, 50 U Benzonase (Millipore), 1x protease inhibitor cocktail (Roche). Protein concentration was measured by DC protein assay kit (Biorad) and 20 µg of protein were separated by 12% SDS-PAGE. The gels were then stained with the Coomassie staining solution (50% Methanol (Fisher Scientific), 10% Acetic Acid (Sigma), 0.1% Coomassie Brilliant Blue R250 (Biorad)) to confirm equal loading. Gels were then dried and exposed to phosphor screens (GE Healthcare). The signal was detected by using Typhoon$^{TM}$ Phosphoimager (GE Healthcare).

**Oxygen consumption rate (OCR) and extracellular acidification rate (ECAR) measurements**. Mitochondrial respiration was assayed in fibroblasts treated or not treated with 2DG on 24-well XF24e plates, using an XF24e Extracellular Flux Analyzer (Agilent Technologies). Briefly, $5 \times 10^4$ cells were seeded approximately 16–24 h before the assay in prewarmed growth medium (DMEM, Gibco) and incubated at 37 °C. Subsequently, the medium was removed and replaced with assay medium (XFBase medium minimal DMEM (Agilent) complemented with 2 mM glucose, 2 mM glutamax and 1 mM pyruvate) and cells incubated for 30 min in a 37 °C non-CO$_2$ incubator. After taking an oxygen consumption rate (OCR) baseline measurement, 1 µM oligomycin, 0.75 µM carbonylcyanide-4-trifluoromethoxy-xyphenylhydrazone (FCCP) and 1 µM rotenone were added sequentially. For the extracellular acidification rate (ECAR) values, the average of the first three measurements of the basal level prior the oligomycin injection was considered.

**Immunofluorescence and DNA labelling**. Control and patients' fibroblasts were grown on chamber slides (ThermoFisher Scientific) and fixed with 4% formaldehyde (Sigma) in PBS (Sigma) for 20 min at 37 °C. After washing, the cells were permeabilized with 0.3% Triton X-100 (Santa Cruz Biotechnologies) in PBS containing 5% FBS. For the 5-bromo-2'-deoxyuridine (BrdU, Sigma) incorporation experiment, the cells were incubated with BrdU 50 µM for 13–16 h, then fixed, permeabilized and treated with HCl 2 N for 20 min at 37 °C. Cells were then blocked with PBS containing 5% FBS and incubated with primary antibody overnight at 4 °C. After washes, slides were incubated with the appropriate secondary antibody for 1 h at room temperature. Slides were then washed and mounted over ProLong® Gold Antifade Reagent (ThermoFisher Scientific) without DAPI nuclear staining.

**Image capture and analysis**. Samples were imaged either on a Leica SP5 TCS Inverted Confocal Microscope or a Nikon Ti Inverted Confocal microscope. The microscope software for Leica was Leica Application Suite X, with the file extension '.lif' format, whereas for the Nikon microscope we used NIS Element Software in a '.nd2' format. Z stack of red, green and blue images was acquired sequentially and merged using ImageJ. Laser power, gain and offset parameters were kept constant for each experiment. The image analysis was performed using the plugins available in Fiji ImageJ (2.0.0-rc-15/1.49h); any adjustments to brightness and contrast were applied linearly to all images in a comparison. Treated cells with far fewer BrdU-positive foci than the corresponding untreated cells, such as those shown in Fig. 2b–d, were scored 'inhibited for mtDNA replication'.

**ATP measurements**. Total intracellular ATP levels were measured by bioluminescence using a luciferin–luciferase system according to the manufacturer's instructions. Cells were plated in duplicate 24-well plates, and treated the following day with 10 mM 2DG or 1 µM oligomycin alone for 24 h, and 10 mM 2DG for 24 h with addition of 1 µM of oligomycin for the last 4 h. One plate was used to determine the total protein amount of samples, and the luminescence signal was normalized to the total amount of protein.

**Mitochondrial membrane potential**. Mitochondrial depolarization was evaluated by measuring the loss of TMRM (tetramethylrhodamine methyl ester; Molecular Probes ThermoFisher Scientific, T668) staining by FACS analysis in non-quenching mode (FACS Analyzer LSRFortessa 5 laser SORP, Becton-Dickinson, Diva Software version 8). Gating strategy is illustrated in Supplementary Fig. 9). Cells were seeded in 12-well plates, treated with 10 mM 2DG or 5TG for 24 h and incubated with 20 nM TMRM and 1.6 µM cyclosporine H (Enzo Life Sciences, ALX-380-286) for 30 min. Cells were washed with PBS, trypsinized, centrifuged, resuspended in 300 µl of PBS and acquired. Then, 1.6 µM oligomycin was added in the test tube to a set of separate untreated samples for 90 min and then acquired. Physical parameters were used to gate the singly dispersed cells. Addition of 4 µM of FCCP (carbonyl cyanide 4-(trifluoromethoxy)phenylhydrazone, Sigma) on untreated samples was used to completely depolarize the mitochondria at the end of the experiment.

**Rate of shift of heteroplasmy**. To compute heteroplasmy shifts β between initial heteroplasmy $h0$ and final heteroplasmy $h$ after time $t$, we used the formula

$$\beta t = \log((h(h0-1))/(h0(h-1))) \text{ from}[8].$$

We used the $z$-test against a null hypothesis of zero mean to compute $P$-values.

**Statistical analysis**. Data were collated in Excel 14.4.8. Statistical analyses were performed using Graphpad Prism (v.7 and 8). Immunoblots and mitochondrial protein synthesis were analysed with Fiji ImageJ. Data were analysed using a two-sided non-parametric Mann-Whitney $U$ test, a two-sided parametric $t$-test, a two-sided one-sample median test and a two-sided Wilcoxon signed-rank test. Multiple comparisons were performed with one-way ANOVA test. Comparisons were considered statistically significant for $P$-values < 0.05 ($^*P < 0.05$, $^{**}P < 0.005$, $^{***}P < 0.001$); ns - not significant. Exact $P$-values are reported in the figure legends. The number of replicates for each independent experiment is stated in the corresponding figure legend.

**Study approval**. The study design complied with all relevant regulations regarding the use of human study participants and was conducted in accordance to the criteria set by the Declaration of Helsinki. Written patient consent was obtained and the study was approved by the Queen Square Research Ethics Committee, London, (09/H0716/76) and NRES Committee North East-Newcastle & North Tyneside 1(16/NE/0267), UK.

**Reporting summary**. Further information on research design is available in the Nature Research Reporting Summary linked to this article.

## Data availability

All analytic data associated with this study are available in the main text, Supplementary information and Source Data file. Source data are provided with this paper.

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

## Acknowledgements

The study was funded by the UK Medical Research Council (intramural award to I.J.H. to 2016, and a Senior Non-Clinical Fellowship to A.S., MC_PC_13029), the Muscular Dystrophy UK (grant 17GRO-PG24-0184-1), the European Commission (MEET Project Grant, 317433). U.F.P. is supported by a predoctoral fellowship from the Basque Government (PRE_2018_1_0253). A.S. receives support from the Brain Research UK. A.S., R.Mc.F., R.W.T., R.D.S.P. and M.G.H. receive support from The Lily Foundation. I.J.H. is supported by the Carlos III Health Program, CiberNED and País Vasco Department of Health (2018111043; 2018222031). I.G.J. receives funds from the European Research Council (ERC) under the European Union's Horizon 2020 research and innovation programme (grant agreement no. 805046, EvoConBiO). R.W.T. and R.Mc.F. are supported by the Wellcome Centre for Mitochondrial Research (203105/Z/16/Z), Mitochondrial Disease Patient Cohort (UK) (G0800674), and the UK NIHR Biomedical

Research Centre for Ageing and Age-related disease award to the Newcastle upon Tyne Foundation Hospitals NHS Trust. R.W.T., R.Mc.F., R.D.S.P. and M.G.H. are supported by the MRC International Centre for Genomic Medicine in Neuromuscular Disease (MR/S005021/1) and the UK NHS Highly Specialised Service for Rare Mitochondrial Disorders of Adults and Children. R.W.T. also receives funding support from The Pathology Society. R.D.S.P. is supported by a Medical Research Council (UK) Clinician Scientist Fellowship (MR/S002065/1).

## Author contributions

A.S. and I.J.H. conceived the study, designed experiments and contributed to data analyses. B.P., D.I., M.M. and D.P.R. contributed to the design of the experiments, performed the bulk of them, and analysed the data. U.F.P., A.L.A., M.M.O. and M.V.F. contributed to study the effects of respiratory inhibitors on mtDNA replication. I.G.J. contributed to the interpretation of the data based on mathematical modelling. L.V. participated in study design and supervised some of the analysis of the Myo.RD cell line. R.D.S.P., R.Mc.F., M.G.H. and R.W.T. provided cell lines and clinical data. A.S. and I.J.H. wrote the paper, with input from all the authors.

## Competing interests

The use of the technology described in this work has been protected in the UK patent GB2116499.1 (Patent applicant, UCL Business Ltd; status pending) with B.P., D.I., I.J.H. and A.S. named as inventors. The other authors declare no conflict of interest.
