## [Peer Review File. · Nature Communications]

2-deoxy-D-glucose couples mitochondrial DNA replication with mitochondrial fitness and promotes the selection of wild-type over mutant mitochondrial DNAREVIEWER COMMENTS

Reviewer #1 (Remarks to the Author):

The manuscript is much improved, and the addition of the glutamine insight is appreciated. The idea that glutamine preferentially fuels OXPHOS in wildtype healthy mitochondria with WT mtDNAs to allow replication is important. As opposed to in mitochondria with mutant mtDNAs and dysfunctional respiratory chains.

However, much of the mechanistic insight supporting their model is correlative. Can the authors demonstrate in cybrid cells that WT mtDNAs are being replicated at a higher rate than mutant mtDNAs upon DG treatment, as opposed to mutant mtDNAs being degraded. Perhaps the authors could take advantage of their BRDU experiment and determine if BRDU is accumulating in WT mtDNA as opposed to mutant mtDNAs upon treatment of the cybrid cells with DG. Reagents are available where BRDU can be immunoprecipitated and the mtDNA interrogated. PMIDs: 26728715, 20150148

Reviewer #3 (Remarks to the Author):

This revised manuscript presents a much-improved logic and narrative, and the current conclusions are now supported by the data. While the manuscript remains descriptive and lacks mechanistic insights, the findings are relevant and worth publishing in this journal. The only request is to determine the heteroplasmy shift (as shown in figure 1) for the experiments proposed in Fig. 5 where the authors only assessed mtDNA replication. It would be important to include in this experiment not only glutamine-free media but also a condition where glutamine metabolism is inhibited, for instance using CB-839. This experiment is vital to corroborate the authors' model whereby glutamine fuels the replication of mutant mtDNA.

Reviewer #4 (Remarks to the Author):

The MS by Pantic, Ives et al. was originally reviewed by 3 Reviewers that had a number of concerns. I was added as a 4th Reviewer to the revised version to assist the Editor assessing the suitability for publication in Nat. Commun.

I found the work interesting, but also had a number of serious concerns.

The most important one, which I consider a "fatal flaw", is the fact that by blocking glycolysis, cells with lower levels of mutant mtDNA would invariably grow a bit better. ATP in culture conditions is essentially provided by glucose. By blocking glycolysis, cells are obliged to use respiration to produce ATP. As quoted "...demonstrating that 2DG targets an important source of non-mitochondrial ATP, i.e. glycolysis to lactate".

The authors argued that preferential growth of cells with reduced mutant mtDNA is not the case because cells grown without glucose did not show decreases in mutant mtDNA. However, this is a false equivalent. 10% FCS has substantial levels of glucose. On the other hand, 2DG and 5TG completely block glycolysis. 2DG does not need to trigger apoptosis. It could promote just small delays in the growth rate of cells with higher levels of mutant mtDNA, thereby increasing the levels of wt mtDNA in the cell population. This passive mechanism is compatible with "the glucose analogues produced a shift rate of 0.38, which would correspond to a change from 50% to 41% mutant mtDNA in one week." Another argument used by the authors is that confluent cells still change heteroplasmy. Again, there is no evidence that there could be a small rate of cell growth. Also, small rates of cell death could contribute for cellular turnover and the changes in heteroplasmy observed. A complete block of cell growth would be required for this experiment to be more informative.

In any case, the mechanism of mutant mtDNA reduction by 2DG would be relevant if it could operate in postmitotic cells. However, respiration deficient postmitotic cells would likely die if glucose utilization is blocked.

The authors also argued that cells tend to preferentially maintain the mutant mtDNA under normal growth conditions. Although arguably, this could be said for postmitotic tissues, it is not true for cultured cells as most mutant mtDNA tend to go down in culture, particularly if the levels of mutant are high.

Essentially all the work flowing from the 2DG or 5TG treatments can be explained by selecting cells with slightly lower mutant mtDNA loads. This includes BrdU labeling.

Another result that strengthened my concerns: "The effect of 2DG on the mutant cells was considerably greater, as it increased the proportion of cells with depolarized mitochondria from 5% to 25% of the total (Fig. 4b)."

Point by point responses to the Reviewers.

(our responses appear in blue font with the original comments of the reviewers in black font)

Reviewer #1 (Remarks to the Author):

The manuscript is much improved, and the addition of the glutamine insight is appreciated. The idea that glutamine preferentially fuels OXPHOS in wild-type healthy mitochondria with WT mtDNAs to allow replication is important. As opposed to in mitochondria with mutant mtDNAs and dysfunctional respiratory chains. However, much of the mechanistic insight supporting their model is correlative. Can the authors demonstrate in cybrid cells that WT mtDNAs are being replicated at a higher rate than mutant mtDNAs upon DG treatment, as opposed to mutant mtDNAs being degraded.

Perhaps the authors could take advantage of their BRDU experiment and determine if BRDU is accumulating in WT mtDNA as opposed to mutant mtDNAs upon treatment of the cybrid cells with DG. Reagents are available where BRDU can be immunoprecipitated and the mtDNA interrogated. PMIDs: 26728715, 20150148

We thank the reviewer for the positive remarks and the excellent suggestion to immunocapture BrdU-labelled mtDNA fragments. We performed the BrdU-immunoprecipitation in mutant cells treated with or without 2DG, and, as predicted, considerably more BrdU-labelled wild-type mtDNA is captured (*versus* mutant) when the heteroplasmic fibroblasts are treated with the compound. As the reviewer will appreciate, this demonstrates that the wild-type mtDNAs have a direct replicative advantage over the mutant molecules in the presence of 2DG (see new **Fig. 3e**, **Supplementary Fig. 3f** and main text).

Reviewer #3 (Remarks to the Author):

This revised manuscript presents a much-improved logic and narrative, and the current conclusions are now supported by the data. While the manuscript remains descriptive and lacks mechanistic insights, the findings are relevant and worth publishing in this journal. The only request is to determine the heteroplasmy shift (as shown in figure 1) for the experiments proposed in Fig. 5 where the authors only assessed mtDNA replication. It would be important to include in this experiment not only glutamine-free media but also a condition where glutamine metabolism is inhibited, for instance using CB-839. This experiment is vital to corroborate the authors' model whereby glutamine fuels the replication of mutant mtDNA.

We appreciate the reviewer's comments and we have now tested the effect of glutamine restriction on the segregation of the mtDNA. Glutamine restriction alone (i.e. with 25 mM glucose) did decrease the mutant load, but was less effective than 2DG; however, no glutamine and low glucose (1 mM) was equal in effect to 2DG (new **Fig. 6**). Given that glutamine inhibits the replication of mtDNA in mutant cells and glucose-fuelled respiration supports the replication of functional mtDNA when both glucose and glutamine are scarce (**Fig. 5**), the results strongly suggest that it is the twin effects of 2DG on glucose and glutamine utilization that enables it to

promote the segregation to wild-type mtDNA. We comment further on glucose in the response to reviewer 4 below.

Reviewer #4 (Remarks to the Author):

The MS by Pantic, Ives et al. was originally reviewed by 3 Reviewers that had a number of concerns. I was added as a 4th Reviewer to the revised version to assist the Editor assessing the suitability for publication in Nat. Commun.

I found the work interesting, but also had a number of serious concerns.

The most important one, which I consider a “fatal flaw”, is the fact that by blocking glycolysis, cells with lower levels of mutant mtDNA would invariably grow a bit better. ATP in culture conditions is essentially provided by glucose. By blocking glycolysis, cells are obliged to use respiration to produce ATP. As quoted “...demonstrating that 2DG targets an important source of non-mitochondrial ATP, i.e. glycolysis to lactate”. The authors argued that preferential growth of cells with reduced mutant mtDNA is not the case because cells grown without glucose did not show decreases in mutant mtDNA. However, this is a false equivalent. 10% FCS has substantial levels of glucose. On the other hand, 2DG and 5TG completely block glycolysis.

We thank reviewer 4 for taking on the task of reviewing the revised manuscript, and his/her comments allow us to make some important clarifications.

Glycolytic ATP production. It is not the case that 2DG and 5TG completely block glycolysis. As direct evidence, in **Supplementary Figure 1 panel b**, ECAR is reduced to 30-40% of control values, and this matches the cellular ATP levels: in controls, 2DG reduces the ATP level 30%, and oligomycin, which fully blocks ATP synthesis in mitochondria, decreases it a further 50%; therefore, the non-mitochondrial (i.e. glycolytic) ATP is 50% of the total and so the residual 20% is equal to 40% of the normal glycolytic ATP production (**Fig. 4a**). Even with a compound like KA that can completely block glycolysis, the effect is dose dependent – that is KA can be applied at a dose that restricts but does not completely block glycolysis.

Serum. As stated in the methods, we used dialyzed FBS that is essentially free of glucose; thus, this is not an issue of concern.

Cell growth. Nevertheless, the reviewer is correct in that the mutant cells have a marked growth disadvantage in 1 mM glucose compared to cells with no mutant. However, this difference disappears when glutamine is withdrawn; and the growth of

the control cells slows dramatically too in glucose restriction without glutamine (Figure below, full set in the **Supplementary Fig. 6**).

Figure. Comparison of the effect of nutrient restriction on the growth of control cells versus those carrying mutant mtDNA. Lowering the glucose concentration from 25 mM (Glc25) to 1 mM (Glc1) slows the growth of cells with mutant mtDNA much more than controls; whereas additionally removing 4 mM glutamine (-Gln) causes near growth arrest of both cell types.

This is exactly what happens with 2DG treatment: the growth of control and mutant cells is arrested equally (see below and **Supplementary Figs. 2 and 6**). We acknowledge this is an important point which we should have drawn attention to in the previous version of the manuscript.

Figure. 2DG inhibits the growth of fibroblasts with mutant or wild type mtDNA. Fibroblasts were seeded at a density of 3×10^4 cells in 6-well plates and imaged every hour. The 2DG treatment was started at 50 or 60% density and the cells were monitored for the following 3 days in an IncucyteTM-adapted incubator. The proliferation rate was determined by analysing the sequence of images with the manufacturer's software to generate growth curves expressing the percentage of cell density over time. At the end of the treatment, the cells were labelled with 5 μ M calcein (Molecular Probes, Thermo Fisher Scientific) for 20 minutes and then imaged (**See Supplementary Fig. 2b**)

2DG restricts glucose and glutamine utilization but not completely, allowing a slow but continuous flux. We had concerns that this would be difficult to reproduce with fixed concentrations in the culture medium. However, as indicated in the reply to reviewer 3, we now show that a combination of glutamine and glucose limitation can reproduce the same increase of wild-type mtDNA as 2DG, in cells that were proliferating prior to the change in growth medium and in contact inhibited cells (**Fig. 6**).

In addition to showing that the 2DG-treated control cells essentially stop growing (**Supplementary Figs 2 and 6 and above**), we have added blots showing that PCNA (Proliferating Cell Nuclear Antigen) levels is strongly repressed in 2DG and contact inhibited cells, as occurs in cells that have exited the cell-cycle (**Fig. 6 and Supplementary Fig. 2d**); and this is evident in control cells after acute 2DG treatment (see below). Hence, there is no evidence of a growth advantage to cells with wild-type mtDNA, whereas there is clear evidence, further substantiated now by the direct test, that 2DG specifically inhibits the replication of mutant mtDNA.

Figure. Control cells stop proliferating after a single pulse of 2DG. Control cells (C2) were treated with 2DG for 48 hours and protein cell extracts analysed for markers of mitochondrial OXPHOS, subunits COX II and NDUF88, and cell proliferation, PCNA. HSP60 and Vinculin (VCL) are indicative of mitochondrial mass and loading, respectively.

2DG does not need to trigger apoptosis. It could promote just small delays in the growth rate of cells with higher levels of mutant mtDNA, thereby increasing the levels of wt mtDNA in the cell population. This passive mechanism is compatible with “the glucose analogues produced a shift rate of 0.38, which would correspond to a change from 50% to 41% mutant mtDNA in one week.”

Having reviewed all our data thoroughly and incorporated new and overlooked data the shift rate has been recalculated as 0.5, but the more important point is the one made above. Namely, that 2DG reduces the growth rate of cells with extremes of mutant or wild-type mtDNA equally. We accept that we could not absolutely exclude there is an indiscernible difference that might possibility account for the slowest rates of segregation seen in some experiments, but it cannot account for the frequently rapid segregation among cells doubling only once a week. Nor can it explain the contact inhibition experiments, see below.

Moreover, while both intracellular and intercellular selection can be expected to attenuate as the mutant load falls and OXPHOS function rises, the recessive nature of the mutant mtDNA means that intercellular selection will always attenuate incrementally, and very slowly plateau around 50% mutant (or above), whereas selection at the organelle level can drive mutant loads considerably lower. We observed decreases in mutant load to 20 and 25% on some occasions, and the mutant load was continuing on a rapid downward trajectory between weeks 6 and 8

of treatment in many other experiments when the analysis was curtailed; none of which is compatible with a 'passive mechanism'.

Another argument used by the authors is that confluent cells still change heteroplasmy. Again, there is no evidence that there could be a small rate of cell growth. Also, small rates of cell death could contribute for cellular turnover and the changes in heteroplasmy observed. A complete block of cell growth would be required for this experiment to be more informative.

Contact inhibition means that the cells (primary fibroblasts) are not growing. As it is achieved when cells are fully confluent, the cells were never passaged and remained in the same wells throughout these experiments. As noted above we have added evidence that 2DG inhibits cells growth both in 'proliferating' and contact inhibited cells (See **Fig. 6 and Supplementary Fig. 2d**).

Figure. Mutant cells do not proliferate in the presence of 2DG. Mutant cells (P2) were treated long-term with 2DG and protein cell extracts analysed for markers of mitochondrial OXPHOS subunits (NDUF8) and cell proliferation (PCNA). HSP60 is indicative of mitochondrial mass.

At the very least we would have expected the segregation to be markedly slower in contact inhibited cells if the true underlying mechanism was INTER-cellular competition; instead, segregation was equally rapid in the contact inhibited cells. As a further test, we performed a new experiment maintaining in parallel cells growing normally prior to 2DG treatment and contact inhibited cells; the shift in the heteroplasmy induced by 2DG was again comparable (**Fig. 6 panel a**).

Cell death. As for cell death, we reiterate that there was no discernible increase in LDH release, and not a single dying cell among the 1000s screened by calcein staining.

In any case, the mechanism of mutant mtDNA reduction by 2DG would be relevant if it could operate in postmitotic cells. However, respiration deficient postmitotic cells would likely die if glucose utilization is blocked.

Here again the reviewer suggests that there either is or is not glucose utilization, whereas all drugs produce dose-dependent responses. As indicated above it was demonstrably not the case that glycolysis was completely blocked by 2DG or 5TG.

While we cannot guarantee that 2DG or 5TG will be equally effective *in vivo*, we explain in the Discussion why the prospects are good. 2DG is an established active pharmacological agent, whose effects in animal disease models are indicative of altered nutrient metabolism. The mitochondrial dysfunction is fundamentally the

same in cells and tissues, and glutamine utilization is elevated in a mouse model of mitochondrial disease, exactly as in the human cell model we have studied; and all our data suggest that stopping cell proliferation (as has occurred in postmitotic tissues) is a requirement to induce the selection of wild-type mtDNA.

The authors also argued that cells tend to preferentially maintain the mutant mtDNA under normal growth conditions. Although arguably, this could be said for postmitotic tissues, it is not true for cultured cells as most mutant mtDNA tend to go down in culture, particularly if the levels of mutant are high.

We agree that this was too broad a statement as it can vary according to the mutant load, the type of mutation and above all, in the light of the new findings, the composition of the growth medium.

Essentially all the work flowing from the 2DG or 5TG treatments can be explained by selecting cells with slightly lower mutant mtDNA loads. This includes BrdU labelling.

Overall the reviewer appears to be arguing that when there is little or no cell growth and no detectable cell death; nevertheless, undetected differences in these parameters can explain the (sometimes extremely rapid and frequently unremitting) segregation to wild-type mtDNA induced by 5TG/2DG to levels well below the threshold for full restoration of OXPHOS function. For the reasons detailed above, we do not regard inter-cellular competition as a credible explanation of the findings. Nor would one expect the process to go into reverse upon the removal of the 2DG, or the addition of rotenone, if we had simply killed or lost the cells in the population with the higher mutant loads.

The reviewer also overlooks the fact that 2DG is unequivocally an INTRA-cellular discriminator at the level of mtDNA replication – all but a few mtDNA molecules in cells with high mutant loads are unable to replicate. This conclusion is strengthened by the new experiment (suggested by Reviewer 1) that demonstrates that the BrdU is preferentially incorporated into WT mtDNAs in the presence of 2DG.

Another result that strengthened my concerns: “The effect of 2DG on the mutant cells was considerably greater, as it increased the proportion of cells with depolarized mitochondria from 5% to 25% of the total (Fig. 4b).”

The increase in the proportion of cells with depolarized mitochondria would only be a concern if it were a surrogate for cell death, which it is not. Note that 2% of cells contain depolarized mitochondria at any one time in a population of healthy control cells. The calcein staining shows that there is no cell death in controls or mutant cells treated with 2DG. Note also that the figure of ‘25% depolarized cells’ was after 48 hours of 2DG treatment. If this was the sub-population of cells that dies or has restricted growth then we would have seen an abrupt drop in the mutant load, followed by no or very slow segregation, which is very different to the actual results.

In summary, our data supporting the hypothesis that 2DG promotes intracellular selection are as follows:

- i) 2DG produces increases in the proportion of wild-type mtDNA while strongly inhibiting cell growth (independent of mutant load) with no detectable cell death.
- ii) Segregation to wild-type occurs equally well in fully confluent cells where cell growth is in abeyance as a result of contact inhibition.

- iii) Segregation to wild-type occurs equally well when an escalating dose regime is applied whereas this should slow the rate of selection if the 2DG favours cells with a lower mutant load.
- iv) 2DG/5TG are able to drive mutant loads down far beyond the 50% threshold level at which OXPHOS is indistinguishable from control cells.
- v) The selection process is reversed when 2DG is removed, whereas if we had killed the cells with the higher mutant loads, or lost them via slower growth, we would have expected the surviving cells to maintain the lower level of mutant mtDNA.
- vi) OXPHOS inhibitors reverse the direction of segregation, consistent with our conclusion that there is a dynamic equilibrium between the selection of mutant and wild-type mtDNA. If selection of the wild-type mtDNA (over mutant) was based on intercellular competition, the inhibitors would negate this advantage but would not favour the mutant; therefore intracellular selection is needed to explain the increase in mutant loads.
- vii) 2DG/5TG inhibit mtDNA replication proportional to the mutant load.
- viii) BrdU preferentially labels wild-type mtDNA in the presence of 2DG, i.e. wild-type mtDNA is synthesized preferentially in the presence of 2DG.

Considering all the above, the weight of evidence is overwhelmingly in favour of intracellular, as opposed to intercellular, selection.

REVIEWERS' COMMENTS

Reviewer #1 (Remarks to the Author):

In general, the authors have addressed my concerns. However, I think the authors may have missed an opportunity by presenting the BrDu as they did.

The authors demonstrate convincingly that DG treatment results in an increased WT/mutant mtDNA ratio. This likely occurs through a preferential increase in WT mtDNA replication as the BrDu immunoprecipitation at 48 hours following DG treatment is enriched in WT mtDNAs. The authors should state explicitly the WT percentage before DG treatment and after 48 hours of DG treatment and compare this to the WT mtDNA percentage labeled with BrDu. Presumably, these data would clearly indicate that BrDu is enriched in WT mtDNAs relative to the input.

Reviewer #3 (Remarks to the Author):

The authors addressed my remaining concerns. Congratulations to the authors for completing this very interesting piece of work.

Reviewer #4 (Remarks to the Author):

The authors have addressed the concerns raised.